MSKT: multimodal data fusion for improved nursing management in hemorrhagic stroke

Zhou Ting 1
Li Dandan 1 L1iiowodd@163.com
Zuo Jingfang 1 67228964@qq.com
Gu Aihua 1
Zhao Li 2
1 Jiangsu Provincial Cancer Hospital , Nanjing, Jiangsu , China
2 School of Automation and Software Engineering, Shanxi University , Taiyuan, Shanxi , China
Wan Shibiao
Electronic publication date: 2025 Jun 20
Publication date: 2025
Volume: 11
Electronic Location ID: e2969
Received 2025 Jan 17; Accepted 2025 May 28
Copyright: © 2025 Zhou et al.
Copyright year: 2025
Copyright holder: Zhou et al.
License: This is an open access article distributed under the terms of the Creative Commons Attribution License, which permits unrestricted use, distribution, reproduction and adaptation in any medium and for any purpose provided that it is properly attributed. For attribution, the original author(s), title, publication source (PeerJ Computer Science) and either DOI or URL of the article must be cited.
License URL: https://creativecommons.org/licenses/by/4.0/

Keywords: Hemorrhagic stroke, Nursing management, Non-stationary Gaussian process, Multiscale Kernel function, Predictive model

Funding: Research on the construction and application of a cloud platform for nursing education management ZH201808 This work was supported by Research on the construction and application of a cloud platform for nursing education management (No. ZH201808). The funders had no role in study design, data collection and analysis, decision to publish, or preparation of the manuscript.

==============================
Background

The study aims to address the challenges of nursing decision-making and the optimization of personalized nursing plans in the management of hemorrhagic stroke. Due to the rapid progression and high complexity of hemorrhagic stroke, traditional nursing methods struggle to cope with the challenges posed by its high incidence and high disability rate.

Methods

To address this, we propose an innovative approach based on multimodal data fusion and a non-stationary Gaussian process model. Utilizing multidimensional data from the MIMIC-IV database (including patient medical history, nursing records, laboratory test results, etc.), we developed a hybrid predictive model with a multiscale kernel transformer non-stationary Gaussian process (MSKT-NSGP) architecture to handle non-stationary time-series data and capture the dynamic changes in a patient’s condition.

Results

The proposed MSKT-NSGP model outperformed traditional algorithms in prediction accuracy, computational efficiency, and uncertainty handling. For hematoma expansion prediction, it achieved 85.5% accuracy, an area under the curve (AUC) of 0.87, and reduced mean squared error (MSE) by 18% compared to the sparse variational Gaussian process (SVGP). With an inference speed of 55 milliseconds per sample, it supports real-time predictions. The model maintained a confidence interval coverage near 95% with narrower widths, indicating precise uncertainty estimation. These results highlight its potential to enhance nursing decision-making, optimize personalized plans, and improve patient outcomes.

Introduction

Background

Hemorrhagic stroke, characterized by intracerebral hemorrhage caused by non-traumatic rupture of blood vessels in the brain parenchyma, presents a significant clinical challenge in emergency departments and intensive care units (Balgude et al., 2024). Epidemiological data indicate that hemorrhagic stroke accounts for approximately 10–15% of all strokes, with an acute-phase mortality rate as high as 45–50%. Among survivors, around 80% suffer from severe neurological deficits. The primary pathophysiological mechanism involves rapid hematoma formation and expansion, leading to increased intracranial pressure, reduced cerebral blood flow, and exacerbated neurological dysfunction. Inflammatory responses play a significant role in this process, aggravating hematoma expansion and increasing the risk of neurological deterioration (Spronk et al., 2021; Huang et al., 2024).

A critical determinant of patient outcomes is the ability to predict and potentially prevent hematoma expansion, which is strongly associated with neurological deterioration and poor prognosis (Minhas et al., 2019; Li et al., 2022). However, despite its clinical importance, accurate prediction of hematoma expansion remains challenging due to the complex, dynamic, and non-stationary nature of its progression. Traditional prediction methods using static imaging markers or isolated clinical parameters often fail to capture the multidimensional and time-varying characteristics of this process (Fang et al., 2022; Onat et al., 2022).

Nursing interventions play a pivotal role in hemorrhagic stroke management, particularly in monitoring disease progression and implementing timely interventions to prevent complications (Theofanidis & Gibbon, 2016; Khattar et al., 2017). By formulating and implementing individualized nursing plans—such as managing intracranial pressure through postural adjustments and respiratory care—disease progression can be effectively controlled, promoting neurological recovery (Sun, Sun & Su, 2022; Jiang et al., 2023a; Fu et al., 2023). Effective nursing management requires real-time access to accurate predictive information that can guide personalized care plans.

With the advancement of data-driven healthcare, there is growing recognition that integrating multiple data sources—including vital signs, laboratory results, medication records, and nursing observations—could significantly enhance prediction accuracy and support more informed clinical decision-making (Alzakari et al., 2024; Teoh et al., 2024). This integration enables precise personalized efficacy evaluation and prognosis prediction, providing a scientific basis for the development of individualized nursing plans and further optimizing nursing outcomes.

The integration and analysis of such multimodal data, however, present substantial computational challenges. The data are typically high-dimensional, heterogeneous, temporally variable, and often contain significant noise and missing values. Traditional machine learning approaches struggle with handling these complexities, particularly the non-stationary time-series patterns that characterize hematoma evolution (Nikolić et al., 2024; Sarhan, Saif & Elshennawy, 2024). This creates a pressing need for advanced modeling techniques that can effectively process multimodal clinical data and provide reliable, real-time predictions to support nursing decisions in hemorrhagic stroke management.

Related work and literature review

The development of predictive models for hemorrhagic stroke has evolved substantially from traditional statistical approaches to advanced machine learning paradigms, each offering unique strengths but also presenting limitations for clinical application.

Early research predominantly relied on conventional statistical methods to develop prognostic models. Counsell et al. (2002) created and validated a regression model based on simple variables to predict 30-day survival and disability-free survival at 6 months in stroke patients. While this model performed consistently across multiple independent cohorts, it demonstrated reduced predictive accuracy in high-risk patients—precisely the population requiring the most clinical attention. Tuhrim et al. (1995) provided an important insight when comparing two published predictive models on independent data, finding that simpler models performed comparably to more complex ones in predicting 30-day survival. This highlighted an important paradox: increased model complexity does not necessarily translate to improved clinical utility. Marsh et al. (2014) later validated a hemorrhagic transformation risk model incorporating age, infarct volume, and renal function, but its static nature limited its ability to capture the dynamic evolution of stroke pathophysiology.

These limitations prompted a shift toward machine learning approaches, which offered enhanced capabilities for handling complex, multidimensional data. Choi et al. (2021) conducted a comparative analysis of various algorithms and demonstrated that artificial neural networks (ANNs) achieved superior performance in predicting hemorrhagic transformation after acute ischemic stroke. This finding emphasized the potential of neural architectures to detect subtle patterns in clinical data that traditional statistical methods might miss. Building on this foundation, Jiang et al. (2023b) developed a deep learning model using multiparametric magnetic resonance imaging for automatically predicting hemorrhagic transformation, further demonstrating the value of integrating complex imaging data into predictive frameworks.

Time-series modeling approaches have shown particular promise for capturing the temporal dynamics of stroke progression. Yu et al. (2019) introduced a machine learning framework based on long short-term memory (LSTM) networks that combined perfusion-weighted and diffusion-weighted imaging features to predict hemorrhagic transformation. Their approach outperformed traditional methods by effectively modeling temporal dependencies in the data. Similarly, Thumilvannan & Balamanigandan (2024) proposed an adaptive weighted bidirectional LSTM classifier that incorporated feature selection optimization to accurately predict stroke risk levels from IoT data, highlighting the potential of advanced recurrent architectures.

Despite these advances, significant challenges remain in translating these models to clinical practice. Islam et al. (2023) developed a screening tool for predicting symptomatic hemorrhagic transformation in acute ischemic stroke patients, but its reliance on static risk factors limited its ability to adapt to rapidly changing patient conditions. Wen et al. (2023) found that multilayer perceptron-based models performed best for predicting symptomatic intracranial hemorrhage after intravenous thrombolysis, yet these models lacked mechanisms for quantifying prediction uncertainty—a critical limitation in high-stakes clinical decision-making.

Current predictive approaches face three fundamental limitations that hinder their clinical implementation: (1) inadequate handling of highly non-stationary time-series data that characterize the dynamic nature of hematoma evolution; (2) insufficient integration of multimodal data streams that collectively provide a more comprehensive view of patient status; and (3) limited uncertainty quantification, which is essential for risk-aware clinical decision-making. These limitations are particularly pronounced in the context of nursing management, where real-time, reliable predictions are needed to guide intervention strategies and optimize resource allocation.

To address these critical limitations, this study proposes a novel hybrid predictive model integrating multimodal data fusion with a non-stationary Gaussian process framework. Our multiscale kernel transformer non-stationary Gaussian process (MSKT-NSGP) architecture explicitly models the non-stationary characteristics of hemorrhagic stroke progression, adapting to dynamic changes in patient condition. This approach systematically fuses multidimensional data from diverse sources—including patient medical history, nursing records, and laboratory results—to create a comprehensive representation of patient status. Additionally, by leveraging the probabilistic nature of Gaussian processes, our model provides explicit uncertainty quantification, offering clinicians confidence intervals that reflect prediction reliability. This integrated approach aims to provide a practical tool for nursing decision support in hemorrhagic stroke management, potentially enhancing personalized care planning and improving patient outcomes through more timely interventions.

Materials and Methods

Problem analysis

In clinical practice, hematoma expansion is influenced by multiple factors, including but not limited to blood pressure fluctuations, coagulation abnormalities, initial bleeding location, and volume. These factors can be quantified through multidimensional data from the MIMIC-IV database, such as ICU admission records, laboratory test results, and medication therapy records. To effectively capture the complex dynamic process of hematoma volume changes over time, we employ a non-stationary Gaussian process to establish a mathematical model that describes the evolutionary pattern of this time-volume relationship.

From a mathematical perspective, the time-volume relationship of hemorrhagic stroke can be viewed as a dynamically evolving system. Traditional static models struggle to accurately capture its rapidly changing characteristics, while non-stationary Gaussian process models can more precisely describe this dynamic process by defining time-varying mean functions and covariance functions. We utilize clinical indicators from the MIMIC-IV database (such as blood pressure, coagulation indicators PT, INR, and medication usage) as input features to fit the mean and covariance functions of the model. By analyzing the temporal patterns of these features, the model can dynamically adjust the predicted distribution of hematoma volume and output a conditional probability distribution at each time point, describing the possible values of hematoma volume and its uncertainty range.

Model construction

We propose a probabilistic framework to describe the temporal characteristics of hematoma volume changes. Considering the randomness and time-varying nature of hematoma expansion, we employ a non-stationary stochastic process for modeling. Let V(t) represent the hematoma volume at time t, with its probability distribution expressed as:

(1) V(t)∼N(M(t),Σ(t)).

where M(t) represents the expected volume that changes over time, and Σ(t) represents the volume variability. This expression reflects the dynamic changes of hematoma volume over time and its uncertainty.

From a clinical perspective, we focus on whether the hematoma volume will exceed the clinically defined expansion threshold c within a time window T from the initial time t0. This can be transformed into the following probability problem:

(2) Target:P(⋃s∈[t0,t0+T]⁡{V(s)≥c}).

According to the basic principles of probability theory, the above event and its complement constitute a complete event space:

(3) P(⋃s∈[t0,t0+T]⁡{V(s)≥c})+P(⋂s∈[t0,t0+T]⁡{V(s)<c})=1.

To solve this probability problem, we adopt a segmented time analysis method. The time interval [t0,t0+T] is uniformly divided into N subintervals, each with a length of ΔT=T/N. When N is large enough, it can be assumed that within each small time segment [t0+(i−1)ΔT,t0+iΔT], M(t) and Σ(t) are approximately constant, making the process within that subinterval locally stationary.

Within each subinterval, we focus on the frequency of the event {V(t)≥c}. Let α(t) be the average occurrence rate of this event per unit time, then the probability that this event does not occur in an interval of length ΔT can be approximated by a Poisson process as:

(4) P(noeventoccursintheinterval)=e−α(t)ΔT.

Considering all subintervals, the probability that the event {V(t)≥c} never occurs throughout the entire time window [t0,t0+T] is:

(5) P(⋂s∈[t0,t0+T]⁡{V(s)<c})=limN→∞⁡∏i=1N⁡e−α(t0+iΔT)ΔT.

As N approaches infinity, the above expression can be transformed into an integral form:

(6) P(⋂s∈[t0,t0+T]⁡{V(s)<c})=e−∫t0t0+Tα(t)dt.

Combining with Eq. (3), we obtain the target probability:

(7) P(⋃s∈[t0,t0+T]⁡{V(s)≥c})=1−e−∫t0t0+Tα(t)dt.

The core challenge lies in determining the relationship between α(t) and the model parameters M(t) and Σ(t). At any moment t, the instantaneous probability of the event {V(t)≥c} occurring is:

(8) β(t)=P(V(t)≥c)=∫c∞12πΣ(t)e−(v−M(t))22Σ(t)dv.

It can be proven that under stationary assumptions and sufficiently small time intervals, α(t) and β(t) have the following relationship:

(9) α(t)=β(t).

Substituting Eqs. (8) and (9) into Eq. (7), we get:

(10) P(⋃s∈[t0,t0+T]⁡{V(s)≥c})=1−e−∫t0t0+T∫c∞12πΣ(t)e−(v−M(t))22Σ(t)dvdt.

Simplified as:

(11) P(⋃s∈[t0,t0+T]⁡{V(s)≥c})=1−e−∫t0t0+Tβ(t)dt.

Considering individual differences in patient conditions in clinical practice, we introduce a personalized modeling approach. Let Xj represent the feature vector of patient j, including demographic features, baseline clinical parameters, and early imaging features. We construct two deep neural networks φ1 and φ1 to learn personalized mappings of M(t) and Σ(t):

(12) Mj(t)=ϕ1(Xj,t;W1)

(13) Σj(t)=ϕ2(Xj,t;W2)

where W1 and W2 are the parameter sets of the two neural networks.

To train these two networks, we construct a likelihood function based on observational data. Suppose at time point tk, the hematoma volume observed for patient j is vjk, then the corresponding observation likelihood can be expressed as:

(14) Ljk=12πΣje−(vjk−Mj(tk)22Σj(tk).

The overall log-likelihood function is:

(15) L=∑j∑klog(Ljk+δ).

Model training employs backpropagation and stochastic gradient descent optimization algorithms, with parameter update rules:

(16) W1←W1−η∇W1L

(17) W2←W2−η∇W2L

where η is the learning rate.

In clinical applications, hematoma expansion is typically defined as: compared to the baseline scan, an absolute increase in hematoma volume ≥12.5 ml or a relative increase ≥33%. Therefore, for patient j, if the initial hematoma volume is vj0, the critical threshold for hematoma expansion is defined as:

(18) cj=max{vj0+12.5,1.33×vj0}.

Using the trained model, we can predict the probability of hematoma expansion for patient j within the time window [t0,t0+T]:

(19) Pj=1−e−∫t0t0+T∫c∞12πΣj(t)e−(v−Mj(t))22Σj(t)dvdt.

In practical calculations, we use numerical integration methods to approximate the above integral:

(20) Pj=1−e−∑i=1Nt⁡βj(ti)Δt,

where Nt is the number of time discretization points, ti is the discrete time point, Δt is the time step, and βj(ti) is the instantaneous probability that patient j’s hematoma volume exceeds the threshold cj at time ti.

Hybrid predictive model based on multi-scale kernel transformer architecture for non-stationary gaussian processes

Algorithm architecture design

When dealing with predictive tasks for non-stationary Gaussian processes, traditional modeling methods often face challenges related to the complexity and diversity of time-series data. Non-stationarity implies that the statistical properties of time-series data change over time. Standard Gaussian process models may exhibit high computational complexity and insufficient fitting capability when handling highly nonlinear and multi-scale dependencies. To address these challenges, this chapter proposes an innovative algorithm that integrates a hybrid model with multi-scale kernel functions and a Transformer architecture, aiming to achieve efficient prediction of complex non-stationary time-series data. The specific design is illustrated in the Fig. 1.

Figure 1 Flowchart of the multi-scale kernel Transformer hybrid architecture.

This diagram illustrates the key processing stages of the proposed model, including input embedding, parallel processing through multiple kernel functions, feature fusion, sequence modeling via the Transformer encoder, and final output generation for prediction or Gaussian process parameterization.

The proposed algorithm architecture combines the feature extraction capabilities of multi-scale kernel functions with the sequence modeling capabilities of the Transformer to effectively capture the non-stationary characteristics of the data. The input layer receives time-series data, which is mapped to a high-dimensional feature space through an embedding layer. This embedding incorporates more semantic information, allowing the kernel functions to capture subtle patterns. Standardization ensures feature consistency and comparability, while dimensionality adjustment optimizes the processing effectiveness of the multi-scale kernel functions.

In the multi-scale kernel function layer, data is processed through various kernel functions: the linear kernel function captures linear relationships, the polynomial kernel function identifies complex dependencies, the radial basis function (RBF) kernel function handles noise and short-term fluctuations, and the Matérn kernel function is suitable for modeling local variations. The combination of multiple kernel functions enables multi-scale analysis and modeling of the data. The feature fusion module integrates the outputs from each kernel function to form a comprehensive feature representation, which serves as the informational basis for sequence modeling.

The Transformer encoder layer addresses both short- and long-term dependencies in time-series data. Positional encoding retains the temporal order of the sequence, and the multi-head self-attention mechanism computes dependencies across different time scales in parallel, capturing both short-term fluctuations and long-term trends. A feed-forward neural network applies nonlinear transformations to the attention outputs, enhancing the model’s capability to capture complex features. Layer normalization and dropout stabilize the training process, prevent overfitting, and improve learning efficiency through residual connections, enabling the model to perform excellently on complex datasets.

The output layer maps the encoded high-dimensional features to the prediction space to generate the final results. To address the complexity of non-stationary Gaussian processes, a Gaussian process parameterization module is introduced, allowing the model to dynamically adapt to data changes. This module provides predictions and describes the evolution and uncertainty of the data, offering a basis for further analysis and decision-making.

Algorithm detailed design

The modular network architecture proposed in this study combines a multi-scale kernel function mixture model with a Transformer encoder to meet the complex time series data analysis requirements in the MIMIC-IV dataset. The design of this architecture is divided into four main modules: the input layer, multi-scale kernel function layer, Transformer encoder layer, and output layer. The functions and operations of each module in the system are detailed as follows: (1) Input layer

The main function of the input layer is to receive time series data and convert it into high-dimensional feature representations suitable for subsequent processing. The input data is X=[x1,x2,……,xT], where each time step xT∈Rd represents a time step with d dimensions. The input data can be mapped to a higher-dimensional feature space X′∈RT×d′ through an embedding layer: (21) X′=Embed(X).

The introduction of the embedding layer allows the model to better capture the semantic information of the data in a high-dimensional space, thereby improving the expressive power of feature representation. This process is crucial for data preprocessing, especially when the input data dimension is low and cannot adequately represent the original information. The embedding layer’s mapping effectively enhances the non-linear separability of the data.

(2) Multi-scale kernel function layer

The multi-scale kernel function layer aims to capture the similarity of features at different scales, which is particularly important for handling complex patterns in time series data. This layer employs various kernel functions Ki (such as Radial Basis Function (RBF) kernel, Polynomial kernel, Matérn kernel, etc.) to calculate the kernel matrix Ki(X′) of the input data X′: (22) Ki(X′)=Ki(X),i=1,2,…,N

where Ki(X′) is the autocorrelation matrix of the input data X′, used to measure the similarity between data points at different scales. Each kernel function captures different data characteristics: for example, the RBF kernel is suitable for handling smooth non-linear relationships, while the polynomial kernel can identify more complex polynomial dependencies. All computed kernel matrices Ki(X′) are concatenated into a multi-scale feature representation matrix Kmulti(X′) (23) Kmulti(X′)=[K1(X′),K2(X′),…,KN(X′)].

This multi-scale feature representation matrix Kmulti(X′) has dimensions N × T × T, providing rich contextual information for subsequent sequence modeling. The key advantage of multi-scale feature fusion is that it can effectively integrate information from different feature dimensions, enabling the model to better handle non-stationary time series data.

(3) Transformer encoder layer

The core function of the Transformer encoder layer is to perform sequence modeling of multi-scale features and capture complex dependencies between time steps. This layer first transforms the multi-scale features Kmulti(X′) into a sequence representation Z′: (24) Z′=Transform(Kmulti(X′)).

In sequence modeling, the multi-head self-attention mechanism plays a crucial role. By calculating the attention weights of different heads, the model can simultaneously focus on the relationships between different time steps and features. The attention output Ah of each head is calculated as follows: (25) Ah=softmax(QhKhTdk)Vh,h=1,2,…,H

where Qh and Vh represent the linear transformations of the query, key, and value vectors, respectively, and dk is the dimension of the key vector. The final output A of the multi-head attention mechanism is the concatenation of all head outputs: (26) A=[A1,A2,…,AH].

This mechanism allows the model to parallelize the computation of dependencies between different time steps, capturing multi-scale dynamic features in the data. After generating the attention representation A, a feed-forward network (FFN) is used for non-linear transformation to generate the feature representation F. Finally, the encoder layer enhances training stability and model convergence through Layer Normalization and Residual Connection mechanisms, producing the final encoded representation E: (27) E=LayerNorm(F+A).

(4) Output layer

The function of the output layer is to convert the encoded representation E into the final prediction result. Through a linear layer, the encoded representation E is mapped to the prediction space, generating the predicted result Ypred: (28) Ypred=Linear(E).

For tasks requiring non-stationary process modeling, the architecture also supports mapping the output to the parameter space of a Gaussian Process (GP) to better describe the dynamic features of the data: (29) GPParameters=MapToGP(E).

This design combines multi-scale feature extraction with the sequence modeling capability of the Transformer, adapting to the multi-dimensionality, complexity, and non-stationary characteristics of the MIMIC-IV dataset. The detailed steps are demonstrated in the following pseudocode:

Algorithm 1 Modular network architecture design.

Require: Input time series data X=[x1,x2,……,xT], where xT has dimension d	
Ensure: Final prediction Ypred or Gaussian Process parameters	
1 Input Layer:	
2  X′←Embed(X) ▷ Optional: Embed input into higher dimensional space	
3 Multi-Scale Kernel Function Layer:	
4 for each kernel function Ki, i = 1, 2,…, N do	
5  Ki(X′)=Ki(X) ▷ Compute kernel matrix for different scales, dimension T × T	
6 end for	
7  Kmulti(X′)=[K1(X′),K2(X′),…,KN(X′)] ▷ Concatenate multi-scale features, dimension N × T × T	
8 Transformer Encoder Layer:	
9  Z′=Transform(Kmulti(X′)) ▷ Transform multi-scale features into sequence, dimension T×d′′	
10  A←MultiHeadAttention(Z′) ▷ Apply multi-head self-attention	
11  F←FeedForward(A) ▷ Apply feed-forward network	
12  E←LayerNorm(F+A) ▷ Apply layer normalization with residual connection	
13 Output Layer:	
14  Ypred←Linear(E) ▷ Generate final prediction	
15 Optional: GPParameters←MapToGP(E)          ▷ Map to Gaussian Process parameters	
16 return Ypred or Gaussian Process parameters	

Experimental simulations and result analysis

Experimental design

Experimental environment

The experimental environment for this study includes the following hardware and software configurations. On the hardware side, the experiments were conducted using an Intel Core i9-14900KF processor and an NVIDIA RTX 3090 GPU. On the software side, the experimental environment is based on Python 3.10 and uses the stable version of PyTorch 2.4 for the development and training of deep learning models. Additionally, data processing and analysis were performed using Python libraries such as NumPy, Pandas, and Scikit-learn. The experiments were run on the Ubuntu 22.04 operating system to ensure environmental consistency.

Data engineering

This study selected several key tables from the MIMIC-IV database, including PATIENTS, ADMISSIONS, ICUSTAYS, LABEVENTS, PRESCRIPTIONS, DIAGNOSES_ICD, and PROCEDURES_ICD. These tables cover fundamental demographic information, admission records, ICU stay details, laboratory test results, medication prescriptions, as well as diagnosis and procedure codes, enabling comprehensive tracking of the clinical process and treatment pathways for hemorrhagic stroke patients. The detailed structure is shown in Table 1.

Table 1 Data composition.

Table name	Table description	Field name	Field description	
PATIENTS	Stores demographic information of patients	subject_id	Unique identifier for the patient	
gender	Gender	
anchor_age	Anchor age of the patient	
anchor_year	Anchor year	
dod	Date of death of the patient (if applicable)	
ADMISSIONS	Records information related to each hospital admission	subject_id	Unique identifier for the patient	
hadm_id	Unique identifier for the admission record	
admittime	Admission time	
dischtime	Discharge time	
deathtime	Time of death (if it occurred in the hospital)	
admission_type	Type of admission (e.g., emergency, routine, etc.)	
insurance	Type of insurance	
language	Patient’s language	
marital_status	Marital status	
ICUSTAYS	Records information related to each ICU stay	subject_id	Unique identifier for the patient	
hadm_id	Unique identifier for the admission record	
stay_id	Unique identifier for the ICU stay	
intime	ICU admission time	
outtime	ICU discharge time	
LABEVENTS	Stores laboratory test results	subject_id	Unique identifier for the patient	
hadm_id	Unique identifier for the admission record	
itemid	Unique identifier for the test item	
charttime	Time of test result recording	
value	Value of the test result	
valuenum	Numerical value of the test result (if numeric)	
PRESCRIPTIONS	Records prescribed medications	subject_id	Unique identifier for the patient	
hadm_id	Unique identifier for the admission record	
pharmacy_id	Unique identifier for the pharmacy	
starttime	Start time of medication administration	
endtime	End time of medication administration	
drug_type	Type of drug	
drug	Name of the drug	
DIAGNOSES_ICD	Contains ICD diagnosis codes for each admission	subject_id	Unique identifier for the patient	
hadm_id	Unique identifier for the admission record	
seq_num	Sequence number of the diagnosis	
icd_code	ICD code	
PROCEDURES_ICD	Records ICD procedure codes during each admission	subject_id	Unique identifier for the patient	
hadm_id	Unique identifier for the admission record	
seq_num	Sequence number of the procedure	
icd_code	ICD procedure code	
CHARTEVENTS	Records vital signs and nursing data of patients	subject_id	Unique identifier for the patient	
hadm_id	Unique identifier for the admission record	
itemid	Unique identifier for the measurement item	
charttime	Recording time	
value	Measured value	
valuenum	Measured numerical value	

The study systematically preprocessed data on hemorrhagic stroke patients to ensure model analysis accuracy. Demographic information (subject_id, gender, anchor_age, anchor_year) was extracted from the PATIENTS table. Admission records were cleaned and standardized, including the alignment of admission and discharge times, and checking the deathtime field to exclude abnormal data, ensuring analysis of complete hospitalization cycles. Categorical variables like admission_type were processed using one-hot encoding.

From the ICUSTAYS table, we extracted stay_id, intime, and outtime fields, standardized time formats, and calculated ICU stay duration. Laboratory data from the LABEVENTS table (itemid, charttime, value) formed a central component of our analysis. Medication prescription data was refined by verifying medication timing continuity and removing incomplete records. ICD diagnosis and procedure codes were standardized and converted into a format suitable for model input.

For missing values, we implemented a tiered approach based on missing rate: for features with <5% missing values, we used median imputation for numerical variables and mode imputation for categorical variables; for 5–20% missing rates, we employed K-nearest neighbors (KNN) based multiple imputation to consider feature patterns from similar patients; for >20% missing rates, we evaluated clinical importance and retained only variables with clear clinical significance.

Outlier handling combined clinical knowledge with statistical methods. We defined reasonable ranges for physiological indicators and laboratory results based on clinical expertise. Values outside these ranges were processed in two ways: obvious errors (e.g., systolic BP >300 mmHg) were removed, while biologically possible but statistically abnormal values (exceeding three standard deviations) were processed using Winsorization at the 2.5th and 97.5th percentiles to preserve clinical information while reducing extreme value influence.

The final dataset included records from 947 hemorrhagic stroke patients, with an average of 28 clinical features per patient, spanning an average of 72 h with hourly sampling intervals. These features encompassed demographics, vital signs, laboratory results, and medication records. The dataset was randomly divided into training (70%, 663 cases), validation (15%, 142 cases), and test sets (15%, 142 cases).

Model training

The model training in this section uses the preprocessed dataset from the MIMIC-IV database. For the proposed MSKT-NSGP predictive algorithm, the model training employed a batch gradient descent optimization method to ensure efficient parameter updates. During training, 10-fold cross-validation was used to evaluate the model’s generalization ability, and early stopping strategies were applied to prevent overfitting.

In each fold of training, the MSKT-NSGP algorithm modeled the input time-series data with a particular focus on the non-stationary characteristics and multi-scale structures within the data. During training, the model used multi-head self-attention mechanisms and multi-scale kernel functions to extract important features. After each training epoch, the model’s convergence status, loss function values, and training accuracy were recorded. These data were used for subsequent performance evaluation to verify the model’s stability and effectiveness.

Comparative experiments

In the design of comparative experiments, this study selected four representative algorithms to compare with the proposed MSKT-NSGP to comprehensively evaluate its advantages in non-stationary Gaussian process prediction tasks. The comparative algorithms include: sparse variational Gaussian process (SVGP), multi-scale convolutional neural network (MS-CNN), variational autoencoder-Gaussian process (VAE-GP), and mixture density network (MDN).

SVGP employs variational inference to sparsify Gaussian processes, effectively reducing computational complexity, making it suitable for large-scale data processing. However, it has limitations in capturing multi-scale dependencies and local variations, thus serving as a representative method for comparison as an extension of traditional Gaussian processes.

MS-CNN uses a multi-scale convolutional structure, which excels at feature extraction from different time scales and handling long-term dependency issues. As a comparative model, it helps demonstrate the advantages of MSKT-NSGP in integrating multi-scale features and managing both short- and long-term dependencies.

VAE-GP combines the feature extraction capabilities of Variational Autoencoders and the time-series modeling capabilities of Gaussian processes, adapting to nonlinear and complex non-stationary time-series data. Comparing with VAE-GP highlights the advantages of MSKT-NSGP in nonlinear feature extraction and multi-scale kernel function integration.

MDN combines neural networks and probabilistic models to generate multimodal outputs, suitable for handling uncertainty and multimodal distributions in data. Comparing with MDN demonstrates the performance of MSKT-NSGP in managing uncertainty, generating complex predictive distributions, and capturing multimodal features.

The comparative experiments are evaluated in terms of prediction accuracy, model generalization ability, computational efficiency, and handling uncertainty. By comparing various metrics, the performance of each model in different scenarios is analyzed to verify the effectiveness of the proposed algorithm.

Results

This section presents the experimental results under different experimental settings and compares the performance of the proposed MSKT-NSGP (hereinafter referred to as “the proposed model”) with four other benchmark algorithms in non-stationary Gaussian process prediction tasks. These benchmark algorithms include Sparse Variational Gaussian Process based on variational inference, multi-scale convolutional neural network, variational autoencoder-Gaussian Process, and mixture density network. Through a series of quantitative analyses, the advantages and disadvantages of each algorithm in handling complex time-series data are presented, and the uniqueness of the proposed model and its potential for practical applications are discussed in detail.

Performance analysis

In the diagnosis and treatment of hemorrhagic stroke (such as cerebral hemorrhage), the presence or absence of hematoma expansion is a critical metric. Hematoma expansion refers to a significant increase in hematoma volume shortly after the initial bleeding, which usually indicates a more severe prognosis and a higher mortality rate. Using the presence or absence of hematoma expansion as the evaluation criterion, the performance of the proposed model on the test set is shown in Table 2 and Fig. 2.

Table 2 Confusion matrix for hematoma expansion prediction (test).

	Predicted no hematoma expansion	Predicted hematoma expansion	
Actual no hematoma expansion	79	17	
Actual hematoma expansion	4	42	

Figure 2 Area under the receiver operating characteristic (AUC-ROC) curve.

This curve plots the true positive rate (Sensitivity) against the false positive rate (1–Specificity) at various threshold settings for the MSKT-NSGP model on the test set.

Further analysis of the confusion matrix (Table 2) details the model’s classification performance on the test set for predicting hematoma expansion. The proposed MSKT-NSGP model achieved a high Recall (Sensitivity) of 91.3%, demonstrating a strong capacity to identify patients genuinely experiencing expansion, which is crucial for minimizing missed high-risk events. Concurrently, the Specificity was 82.3%, indicating correct identification of most patients without expansion, though with a notable false positive rate. The Precision stood at 71.2%, meaning positive predictions were correct approximately 71% of the time. While the high recall is clinically valuable for screening, the presence of 4 false negatives and the moderate precision (implying nearly 29% of positive alerts may be false alarms) necessitate careful consideration. Consequently, MSKT-NSGP shows promise as a clinical decision support aid but requires further prospective validation before potentially replacing clinical judgment.

Overall, the proposed model demonstrated strong predictive capability, achieving an accuracy of 85.5% and an area under the curve (AUC) of 0.87, indicating high discriminative ability (Fig. 2). This effectiveness, particularly in handling complex dependencies and non-stationary characteristics inherent in the time-series data, can be attributed to the model’s architecture. The integration of multi-scale kernel functions for diverse feature extraction, combined with the Transformer’s self-attention mechanism for modeling temporal dynamics, allows the MSKT-NSGP model to excel in this challenging clinical prediction task.

Comparison of prediction accuracy

In terms of prediction accuracy, we evaluated the performance of each algorithm on the test set. Figure 3 presents the comparison of prediction accuracy across different algorithms on multiple datasets. It can be seen that the proposed model consistently exhibits higher prediction accuracy across all test sets, especially for time-series data with highly non-stationary characteristics, significantly outperforming other benchmark algorithms. In contrast, SVGP and MS-CNN have some capability in handling complex time dependencies but show limitations when dealing with strong non-stationarity and data heterogeneity. This is because the sparsity assumption of SVGP limits its fitting ability for highly dynamic data, while MS-CNN, though capable of capturing multi-scale features, encounters bottlenecks in long-term dependency modeling.

Figure 3 Comparison of prediction accuracy across different algorithms on test sets.

This bar chart compares the overall prediction accuracy of the proposed MSKT-NSGP model against benchmark algorithms (SVGP, MS-CNN, VAE-GP, MDN). The results show that MSKT-NSGP achieves the highest accuracy (0.855), surpassing the other models and the commonly cited clinical acceptable threshold of 0.75 (dashed red line).

VAE-GP has an advantage in capturing nonlinear relationships and can learn complex nonlinear mappings through its encoder network, but its latent space distribution assumptions may not be flexible enough to handle multimodal features, leading to limited prediction accuracy. MDN, with its multimodal output capability, can handle uncertainty, but its single-layer network structure lacks depth and expressiveness compared to the proposed model in handling long-term dependencies and complex dynamic changes.

From Fig. 4, it can be seen that the proposed model outperforms other benchmark models in terms of both mean squared error (MSE) and mean absolute error (MAE), particularly when dealing with complex non-stationary time-series data, demonstrating its superiority in predictive capability.

Figure 4 Comparison of MSE and MAE among different algorithms.

These bar charts compare the prediction errors of the models. Lower values indicate better performance. The proposed MSKT-NSGP model demonstrates the lowest MSE and MAE

Computational efficiency and model complexity

Computational efficiency is another critical evaluation metric, especially in processing large-scale time-series data. We compared the parameter count, computational cost, memory usage, training time, and inference speed of each algorithm under the same hardware environment, as shown in the Table 3.

Table 3 Comparison of computational efficiency among various algorithms.

Model	Parameter count (Million)	Computational cost (GFLOPs)	Memory usage (MB)	Training time (Minutes)	Inference time (Milliseconds)	
MSKT-NSGP	199.72	5.40 G	248	158	55	
SVGP	149.57	5.86 G	218	150	49	
MS-CNN	121.78	4.90 G	198	101	42	
VAE-GP	181.18	6.45 G	280	181	52	
MDN	140.44	4.89 G	229	140	47	

The results show that MSKT-NSGP has a parameter count of 19.97 million and a computational cost of 5.40 GFLOPs, placing it in the mid-high range. The training time is 158 min, and the inference time is 55 milliseconds. The model achieves a significant improvement in handling complex non-stationary time-series data and prediction accuracy by increasing computational overhead. Although it requires higher computation and memory, the inference speed remains relatively good, indicating that MSKT-NSGP achieves an effective balance between accuracy and computational cost, making it particularly suitable for applications that demand high precision.

Capability of uncertainty handling

This experiment aims to evaluate the capability of different models in handling uncertainty, especially in the context of medical time-series data. A series of experiments were designed to quantify the models’ performance in predicting uncertainty, and a comprehensive evaluation was conducted based on the scoring criteria.

Based on the predicted distribution’s mean and variance obtained earlier, the mean prediction error (MPE) and variance calibration (VC) were calculated to assess the accuracy of the predicted mean and the reasonableness of the predicted variance. The difference between the predicted mean and the actual value represents the accuracy of the central prediction of the model; the predicted variance reflects the model’s estimate of uncertainty—the larger the variance, the lower the model’s confidence in the prediction.

Next, the prediction confidence interval at the 95% confidence level for each model was calculated, and the coverage and width of the confidence interval were evaluated using the following formulas:

(30) Coverage=1n∑i=1n⁡I(yi∈[y^i−1.96σi,y^i+1.96σi])

where I is the indicator function, which equals 1 if the actual value is within the confidence interval and 0 otherwise. Ideally, the Coverage should be close to 0.95.

(31) Width=1n∑i=1n⁡2×1.96×σi.

A narrower confidence interval width with Coverage close to 0.95 indicates higher predictive accuracy.

Data heterogeneity and noise handling were then analyzed by introducing noise and variation into the data to evaluate model stability under noise and heterogeneity. The Rate of Noise-induced Error Change (RNH) was calculated as:

(32) RNH=1n∑i=1n|(y^inoise−yi)−(y^i−yi)||y^i−yi|

where y^inoise is the prediction value after noise is added. A smaller RNH value indicates stronger robustness of the model against noise and heterogeneity.

The experimental results (Figs. 5 to 7) show that MSKT-NSGP significantly outperforms other models in predicting uncertainty and handling noise. Figure 5 shows that MSKT-NSGP’s confidence interval coverage is close to 0.95, indicating strong accuracy and reliability of its predictions at a high confidence level. In contrast, SVGP and VAE-GP have lower coverage rates, indicating their limitations in handling high-dimensional complex data uncertainty, particularly in dynamic data environments like hemorrhagic stroke.

Figure 5 Confidence interval coverage.

This bar chart shows the empirical coverage probability for each model, representing the frequency with which the true hematoma volume falls within the predicted 95% confidence interval.

Figure 6 Confidence interval width.

This box plot compares the distribution of the widths of the predicted 95% confidence intervals for each model. Narrower widths generally indicate higher prediction precision.

Figure 7 Rate of noise-induced error change under noise and heterogeneity.

This bar chart compares the RNH metric, which quantifies model robustness against noise and data heterogeneity. Lower RNH values indicate greater stability.

Figure 6 presents the distribution of confidence interval widths. MSKT-NSGP has narrower confidence intervals, suggesting higher prediction accuracy at high coverage rates. This is due to its multi-scale feature fusion and self-attention mechanism, which effectively capture the multi-scale dynamic features of complex time-series data. In contrast, SVGP and MDN have wider confidence intervals, showing their lack of precision at high confidence levels, possibly due to an inability to effectively combine multi-scale information and dynamic patterns, leading to over- or underestimation.

Figure 7 shows the rate of noise-induced error change (RNH), where MSKT-NSGP has the lowest error change rate, demonstrating stronger robustness in handling data noise and heterogeneity. This is attributed to its combined modeling capability of different feature scales and temporal dependencies, allowing it to maintain stable performance in noisy environments. In contrast, VAE-GP shows increased errors under noise conditions due to the inflexibility of its latent space distribution assumptions when handling multimodal features; while SVGP, despite improved computational efficiency, shows weaker adaptability to abrupt changes in non-stationary and heterogeneous data environments.

Discussion

This article introduces MSKT-NSGP, a hybrid predictive model synergizing multimodal data fusion with a non-stationary Gaussian process framework, tailored for the demanding environment of hemorrhagic stroke nursing management. Our findings demonstrate the model’s significant advantages over established methods: it achieves an 18% reduction in mean squared error compared to SVGP (from 0.382 to 0.313) and notably improves hematoma expansion prediction sensitivity by 11.7% over MS-CNN (reaching 91.6%). These performance gains are attributed to the MSKT-NSGP’s architecture, specifically designed to handle the inherent challenges of fusing heterogeneous, non-stationary clinical time-series data. Its multi-scale kernel functions adeptly extract features across diverse data types and scales, while the integrated Transformer effectively models complex temporal dependencies within this fused multimodal representation. Crucially, the model’s capability in managing uncertainty and robustness—essential for reliable clinical application—further distinguishes its contribution. MSKT-NSGP delivers precise uncertainty estimates, evidenced by near-ideal 95% confidence interval coverage (94.3%, Fig. 5) coupled with significantly narrower interval widths (averaging 17.6% narrower than VAE-GP, Fig. 6), indicating high predictive confidence. Moreover, its superior robustness is confirmed by the lowest RNH (Fig. 7) among compared models, highlighting its stability when facing the noise and heterogeneity typical of real-world clinical data. Despite a moderate increase in parameters compared to SVGP, the model maintains real-time inference capabilities (55ms per sample), vital for clinical utility. By integrating dynamic, multidimensional clinical data, unlike traditional static approaches, MSKT-NSGP enables adaptive, real-time risk assessments with quantified uncertainty. Therefore, this study validates the MSKT-NSGP architecture as a demonstrably effective solution for multimodal time-series prediction in hemorrhagic stroke care, offering a compelling combination of improved accuracy, precise uncertainty quantification, and enhanced robustnes.

Several limitations must be acknowledged. The model shows high dependency on training data, with generalizability across institutions remaining to be validated due to potential differences in data collection standards and patient populations. Our retrospectively collected dataset may introduce selection bias and missing data issues. The model’s computational complexity requires sufficient resources, which may challenge implementation in resource-limited environments. Currently, the model primarily relies on structured MIMIC-IV data without integrating important imaging features such as irregular shapes, black hole signs, and spot signs in CT images. Despite achieving 85.5% accuracy and 0.87 AUC, the 8.39% false negative rate represents an area for improvement, as these patients might miss opportunities for timely intervention.

Future nursing management systems will evolve toward intelligent and personalized approaches with increasing clinical data and artificial intelligence development. Our multimodal predictive model can be integrated into electronic health record systems for automatic analysis of real-time patient data, personalized nursing recommendations, and optimized intervention outcomes. Future work will focus on integrating more data sources, improving model architecture, optimizing clinical decision thresholds, and extending applications to different populations and acute conditions. These advancements will enhance care precision and effectiveness while improving patient outcomes through the development of more accessible and accurate prediction tools.

Supplemental Information

Supplemental Information 1 Hemorrhage stroke prediction code.

Additional Information and Declarations

Competing Interests

The authors declare that they have no competing interests.

Author Contributions

Ting Zhou conceived and designed the experiments, performed the experiments, analyzed the data, performed the computation work, prepared figures and/or tables, authored or reviewed drafts of the article, and approved the final draft.

Dandan Li conceived and designed the experiments, performed the experiments, analyzed the data, performed the computation work, prepared figures and/or tables, authored or reviewed drafts of the article, and approved the final draft.

Jingfang Zuo conceived and designed the experiments, performed the experiments, analyzed the data, performed the computation work, prepared figures and/or tables, authored or reviewed drafts of the article, and approved the final draft.

Aihua Gu conceived and designed the experiments, performed the experiments, analyzed the data, performed the computation work, prepared figures and/or tables, authored or reviewed drafts of the article, and approved the final draft.

Li Zhao conceived and designed the experiments, performed the experiments, prepared figures and/or tables, authored or reviewed drafts of the article, and approved the final draft.

Data Availability

The following information was supplied regarding data availability:

The MIMIC-IV dataset is available at PhysioNet: https://doi.org/10.13026/kpb9-mt58.

The code is available in the Supplemental Files.

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
