# Peer review of "MSKT: multimodal data fusion for improved nursing management in hemorrhagic stroke"

_PeerJ Computer Science, doi:10.7717/peerj-cs.2969_

## Round 0.1 · original submission · Major Revisions

· Academic Editor

Major Revisions

The reviewers have substantial concerns about this manuscript. The authors should provide point-to-point responses to address all the concerns and provide a revised manuscript with the revised parts being marked in different color.

**Language Note:** The review process has identified that the English language must be improved. PeerJ can provide language editing services - please contact us at [email protected] for pricing (be sure to provide your manuscript number and title). Alternatively, you should make your own arrangements to improve the language quality and provide details in your response letter. – PeerJ Staff

Reviewer 1 ·

Basic reporting

-

Experimental design

I recommend adding some details of data pre-processing to the Data Engineering subsection. How were the missing values and outliers exactly treated?

I think the dataset dimensions should be also added to article. How many observations (patients), features and time points did the data contain?

Validity of the findings

The confidence intervals for the metrics compared in Figures 3, 4, and 7 should be estimated and shown on the barplots.

Using a confusion matrix (see Table 2), estimate the precision, specificity, and recall (sensitivity) for the positive class (the presence of hematoma expansion). Add the values obtained to “Performance analysis”. Are they appropriate to apply the model proposed in clinics?

Additional comments

I advise removing the subsection “Fundamental research on hemorrhagic stroke”. Firstly, the research considered is about the modeling of time series data with machine learning approaches. The findings can potentially be converted into better treatment of stroke patients. But the article is not about the stroke, its risk factors, pathogenesis and clinical appearances. Secondly, this subsection is just a list of facts related to hemorrhagic stroke. It does not help to understand the importance of the proposed solutions in the article. It is a waste of the reader's time.

The same goes for the subsection “Development of nursing in hemorrhagic stroke”. I think the Background section has enough information on this topic. The main result of this research is the computational model that predicts the size of a hematoma. The authors did not directly solve the problems of nursing under hemorrhagic stroke. Did they provide an innovative nursing plan? No. If so, there is no need to dilute quite interesting research with information that is not directly connected to the findings made in the research. Focus on the main topics. Less is better.

I have concerns about the applicability of the findings made in this research. The patient was delivered to the hospital and diagnosed with an intracranial hemorrhage. At the beginning of hospitalization, there is little information about him or her. What kind of clinical data should be provided to predict the expansion of the hematoma in this case? I am afraid the lack of such information restricts the applicability of the solution presented in the article.

The Discussion section needs some details. I advise adding some numbers showing how the model outperforms the other models.

The Discussion also lacks the pros and cons of the proposed solution. I recommend adding a paragraph that discusses the limitations of the method applied.

Reviewer 2 ·

Basic reporting

-

Experimental design

-

Validity of the findings

-

Additional comments

1. The technical explanation of the non-stationary Gaussian process model (Line 242-296) is overly complex and poorly structured. The transition between concepts is abrupt, making it difficult for readers without specialized statistical knowledge to follow. The overall methodology section contains many technical terms without adequate explanation of why these specific components were selected.

2. The literature review is a bit disorganized, reads like a collection of summaries rather than a critical analysis. The paper would benefit from thorough editing for clarity, conciseness, and precision in language.

3. Why are there only three data points for the AUC-ROC curve in Fig. 2? Is it reliable to obtain an AUC of 0.87 with such sparse data points?

4. For all figures, additional caption explanations are needed to improve clarity.

5. The paper claims novelty in applying multimodal data fusion to hemorrhagic stroke nursing, but the figures of prediction accuracy (Fig. 3), MAE (Fig. 4), and Coverage (Fig. 5) did not differentiate this approach from previous work in the field, making the contribution unclear.

---

## Round 0.2 · accepted · Accept

· Academic Editor

Accept

Reviewers are satisfied with the revisions, and I concur to recommend accepting this manuscript.

Reviewer 1 ·

Basic reporting

no comment

Experimental design

no comment

Validity of the findings

no comment

Additional comments

The authors have carefully addressed all recommendations made by the reviewer and substantially revised the manuscript. In my opinion, it has improved considerably and now better meets the journal’s requirements. I wish the authors a success in further development of their research and support the publication of this article.

Reviewer 2 ·

Basic reporting

Reads well.

Experimental design

Reads well.

Validity of the findings

Reads well after the revision.